# Anti-Inflammatory and Anti-Adipogenesis Effects of *Alchemilla vulgaris* L., *Salvia officinalis* L., and *Vitis vinifera* L. in THP-1-Derived Macrophages and 3T3-L1 Cell Line

Bayan Mansour [1], Nora Shaheen [2], Abdalsalam Kmail [1], Nawal Haggag [2], Salah Saad [1], Omar Sadiq [1], Ramez Zaid [1] and Bashar Saad [1,*]

[1] Qasemi Research Center, Al-Qasemi Academy, Faculties of Medicine and Arts and Sciences, Arab American University Jenin, Jenin P.O. Box 240, Palestine
[2] Faculty of Women for Arts, Science and Education, Ain Shams University, Cairo P.O. Box 11566, Egypt
* Correspondence: bashar.saad@aaup.edu or bashar@qsm.ac.il

**Abstract:** The production of pro-inflammatory and anti-inflammatory cytokines, as well as adipocyte differentiation and fat accumulation in the 3T3-L1 mouse embryo fibroblast cell line and the human monocytic cell line THP-1 were measured to determine the anti-inflammatory and antiadipogenic effects of ethanolic extracts of verjuice (unripe grape juice (*Vitis vinifera* L.)), *Salvia officinalis* L., and *Alchemilla vulgaris* L. On both cell lines, the three extracts had much greater cytostatic effects than cytotoxic effects. With an IC50 of 505 μg/mL, *S. officinalis* had the highest cytostatic effect on THP-1-derived macrophages. After treatment with 125 μg/mL, the three extracts dramatically reduced the LPS-induced NO generation in THP-1-derived macrophages from 80 μM to control values after treatment with 125 μg/mL. Furthermore, the extracts reduced the levels of TNF-α and IL-6 production in a dose-dependent manner with the highest effects reached at 250 μg/mL. The production of TNF-α decreased at higher levels compared to IL-6 production. *V. vinifera*, *S. officinalis*, and *A. vulgaris* extracts improved the production levels of IL-10 from 32 pg/mL to 86 pg/mL, 98 pg/mL, and 80 pg/mL at an extract concentration of 125 μg/mL, respectively. The adipocyte differentiation and fat accumulation in 3T3-L1 were decreased to 20% of control values after treatment with plant extracts. Taken together, these results suggest that *V. vinifera*, *S. officinalis*, and *A. vulgaris* likely exert their anti-obesity effects through cytostatic actions and modulation of pro-inflammatory and anti-inflammatory cytokine production, as well as by reducing adipocyte differentiation and fat accumulation.

**Keywords:** anti-inflammatory; anti-adipogenic; obesity; diabetes; cytostatic effects

## 1. Introduction

Obesity, characterized by excessive accumulation of fat in the body (body mass index [BMI] $\geq$ 30 kg/m$^2$), is a complex, chronic, multifactorial disorder which is related to many health issues independently or in association with other factors. These include insulin resistance, hypertension, diabetes type II, hyperlipidemia, coronary heart diseases, respiratory complications, and increase incidence of certain types of cancer [1,2]. Obesity results from adipose tissue expansion by adipocyte hypertrophy or adipocyte hyperplasia or a combination of both which leads to excessive lipid storage [3]. Adipogenesis, the differentiation of preadipocytes to mature lipid-storing adipocytes, consists in general of two main steps: preadipocyte replication and differentiation [4,5]. Adipogenic transcription factors SREBP-1C, PPAR-γ, and C/EBP-α are considered key regulators of adipogenesis [6–8].

Inflammation is now well recognized as a major contributor to obesity, type II diabetes, and insulin resistance. It is characterized by high acute-phase reactants (APR), aberrant synthesis of inflammatory mediators, activation of inflammatory signaling pathways, and

other mediators [9]. Increased weight gain alters the adipose tissue's normal cell composition as a result of macrophages' infiltration, which is brought on by tissue damage brought on by anoxia, apoptosis, or necrosis within the quickly expanding adipose tissue [10]. The presence of macrophages raises the levels of inflammatory cytokines like tumor necrosis factor-alpha (TNF-$\alpha$), monocyte chemoattractant protein-1 (MCP-1), and interleukin-6 (IL-6) [11,12]. This increase in cytokine production results in developing a low-grade chronic inflammation within the adipose tissue which promotes the development of the metabolic syndrome and increases obesity creating an infinite loop [13].

Obesity is treated using anti-obesity medicines such as orlistat, sibutramine, and topiramate [14]. These medications are associated with adverse effects, including dry mouth, anorexia, sleeplessness, and gastrointestinal irritations [15]. Due to these adverse side effects, recent drug trials have focused on herbal-based medicines [16]. Herbs have been used as traditional, natural medicines for healing many diseases due to their potential effects in improving and maintaining human health, low side effects, and low cost. Particularly, various traditional medicinal herbs are stated to have biological impacts [17].

Verjuice (the unripe grape juice of *Vitis vinifera* L.), *S. officinalis* L., and *A. vulgaris* L. are very popular traditional and modern medicinal plants used worldwide [18–23]. The use of unripe grape juice in cooking and medicine has a long history [23]. For example, Hippocrates of Kos mentioned the use of unripe grape juice in treating ulcers and as a digestive agent. Muhammad Mu'min back in 1669 recommended in his famous book "Tuhfat al-Muminin" the use of verjuice as a digestive agent after the ingestion of fatty foods [24]. Verjuice, the unfermented juice acquired from unripe grape berries by pressing [21], is characterized by high acidity, low sugar content, and sour taste. Unripe grape or sour grapes, known as Hosrum in Palestine, is used as a popular acidifying and flavoring agent for many foods and drinks such as salads and soups [22]. It is also widely used in Iran where it is called "Abe ghure" (persian: unripe grape water) and Turkey where it is called "Koruk suyu" (Turkish: unripe grape juice) in cuisine and as a digestive drink [25].

*S. officinalis* L. (Sage) is a shrubby perennial plant belonging to the Lamiaceae family. Its aerial components have long been employed in traditional medicine and cooking. in European folk medicine, it has been utilized to treat inflammations, age-related cognitive disorders, excessive sweating, and dyspepsia [26].

*A. vulgaris* L. (Lady's mantle or Lion's foot), is a perennial herbaceous plant that grows wildly in western Asia, Europe, and North America, especially on verges and upland grassland, and belongs to the Rosaceae family [19]. *A. vulgaris* is utilized extensively in traditional medicine to heal abscesses, nose bleeding, mucous membranes inflammations in the mouth in addition to various gynecological, obesity, and gastrointestinal diseases [27,28].

The current study seeks to assess the effects of three traditionally used anti-obesity medicinal plant extracts on cytokine production, adipocyte differentiation, and fat accumulation in mouse embryo fibroblasts (3T3-L1) and human monocytic cell line THP-1-derived macrophages.

## 2. Material and Methods

### 2.1. Plant Extract

2.1.1. Verjuice

Unripe grapes of *V. vinifera* L. were collected in the early morning in mid-July from Burqin, Jenin, West Bank, Palestine. The berries from the unripe grapes were picked from bunches and thoroughly washed with tap water, fresh juice was prepared by crushing 1.479 kg of the whole berries using a glass jar food blender (MOULINEX, LM435127, France). The mixture was then run through a plastic sieve (size of the mesh sieve pore is 0.5 mm) into a glass beaker at 0–4 °C to slow down the oxidation process. One liter of the juice was collected, then the juice was filtered using a 0.22 μm filter and pH of the juice was determined using a pH meter at room temperature (pH 2.60). Finally, the extract was divided in to 50 mL falcon tube aliquots and stored immediately at −20 °C until use [29].

### 2.1.2. *S. officinalis* and *A. vulgaris*

The aerial parts of *S. officinalis* and *A. vulgaris* plants were harvested in the early morning during July 2019 from the Jenin region, Palestine. The collected plant parts were washed and placed to dry in shade. Forty-five grams of the air-dried powdered plants were placed in a 500 mL Erlenmeyer flask along with 300 mL of 1:1 ethanol/water then transferred to a 70 °C water bath for about 15–20 min to give a dark green extract. Extracts were filtered using medical gauze and squeezed manually by hand in aseptic conditions then centrifuged twice at 3000 rpm for 15 min. The stock extracts were preserved in a falcon tube at −20 °C until use.

### 2.2. Cell Viability Tests

THP-1 and 3T3-L1 cell lines were cultivated and maintained in accordance with earlier research that has been published [30]. The succinate dehydrogenase enzyme changes the water-soluble yellow 3-(4,5-dimethylthiazol-2-yl)-2,5-diphenyl tetrazolium bromide (MTT) into a purple water-insoluble formazan crystal, which allows for a quantitative measurement of live cells [31].

Cells were seeded in a 96-well plate and cultured at 37 °C for 24 h before being treated with varying doses of the produced plant extracts. After discarding the treatment solution, 100 μL of (0.5 mg/mL) MTT was added to the treated cells, which were then incubated at 37 °C for 4 h. After 4 h, the MTT solution was discarded, 100 μL of (9:1) isopropanol alcohol and formic acid were added, and the formazan crystals were eluted for 15 min at room temperature away from light. In an ELISA reader, the optical density (OD) of the MTT formazan was measured at 570 nm, and the proportion of absorbance of treated cells relative to control (untreated) cells was used to calculate cell viability, with absorption of control set to 100%.

#### 2.2.1. Cytotoxic Effects

In a 96-well plate, 100 μL of ($2.0 \times 10^4$ cells/well) suspension was seeded and incubated for 24 h at 37 °C. Following that, cells were treated with various doses of plant extracts and incubated for 24 h at 37 °C to perform the MTT assays.

#### 2.2.2. Cytostatic Effects

The cells were seeded at a lower density in each well ($1.0 \times 10^4$ cells/well) then exposed to the precise doses of plant extract used in the cytotoxicity test and incubated for 72 h at 37 °C to perform the MTT assay and evaluate anti-proliferative activity.

### 2.3. NO Production

By converting nitrite into a purple-colored azo product that can be detected using a spectrophotometer at 550 nm, Griess reagent is used to evaluate nitric oxide levels in vitro [32].

For 24 h, Sigma-Aldrich, (St. Louis, MI, USA) Phorbol-12-Myristate-13- Ac-etate (PMA) (100 ng/mL) and Vitamin D3 (0.1 M) (Sigma-Aldrich, St. Louis, MI, USA) were used to induce the differentiation of THP-1 cells. THP-1 cells were then seeded in a 96-well plate ($1.0 \times 10^4$ cells/well) for 72 h after being stimulated with bacterial lipopolysaccharides (LPS) at a concentration of 5 μg/mL as well as different extract concentrations. A total of 50 μL of supernatant and 200 μL of Griess reagent were added to an empty 96-well plate, incubated at room temperature for 20 min, and then the absorbance at 550 nm was measured using an ELISA plate reader.

### 2.4. Adipogenicity Assessment
#### 2.4.1. 3T3-L1 Differentiation

Before the cells reached complete confluence (day 0), the media was replaced with an induction medium containing 500 mM 3-isobutyl-1-methylxanthane (IBMX), 1 mM dexamethasone, and 1 μg/mL insulin for two days. Every two to three days for a total of

ten days, the induction medium was switched out for maintenance medium, which only included 1.5 mg/mL of insulin and DMEM medium with 10% FBS [33].

### 2.4.2. Effect of Plant Extracts on Lipid Accumulation and Adipogenesis on 3T3-L1 Preadipocyte

For the evaluation of adipogenesis, cultured cells were exposed to different doses of plant extracts (0–500 mg/mL) coupled with differentiation induction medium and maintenance medium from day 0 to day 10. From day 10 to day 15 for the assessment of lipid accumulation, extracts were added to fully differentiated adipocytes. Oil Red O was used to stain the cells, and 200× and 400× magnification images were taken using an inverted microscope (Olympus, Inverted microscope L0390117 supported with Optika, Microscope camera F0480255, Japan).

### 2.4.3. Oil Red O staining

Oil Red O powder (O0625, Sigma-Aldrich Company, St. Louis, MI, USA) was used to make the stock solution, which was then filtered via a 0.22 µ filter. The Stock solution was diluted to 2:3 with distilled water to make 6 mL of working solution, which was then filtered through a 0.22 m filter and allowed to sit for 30 min at room temperature.

Cells were cultivated in 96-well plates, followed by three PBS washes, a 15-min fixation in 4% formaldehyde in PBS, and two rinses in distilled water. The cells were then stained for 30 min with a freshly prepared working solution (0.132 mL/cm$^2$), rinsed three times with distilled water, and the remaining dye was removed from the cells using 100% isopropanol (0.263).

### *2.5. Determination of TNF-α, IL-6, and IL-10 Production*

TNF-α, IL-6, and IL-10 release were measured in the supernatants of LPS-activated THP-1-derived macrophages using an ELISA commercial kit (available from Sigma-Aldrich, St. Louis, MI, USA) after 4 h, 6 h, and 24 h of treatment, according to the manufacturer's protocol. The standard curve was used to determine cytokine levels, which were then expressed in pg/mL.

### *2.6. Statistical Analysis*

Results and values from each experiment were carried out in triplicate, and they are presented as mean and SD. Using GraphPad PRISM 8, a one-way ANOVA was used to perform multiple comparisons, and then Dunnett's test was applied. The threshold for significance was set at $p < 0.05$.

### 3. Results and Discussion

### *3.1. Cytotoxic and Cytostatic Effects*

Here, we used the MTT assay to test the inhibitory effects of the extracts of verjuice, *S. officinalis*, and *A. vulgaris* on 3T3-L1 cells and THP-1-derived macrophages after 24 h (cytotoxic effects) and 72 h (cytostatic effects) treatment with increasing doses of the three plant extracts. Verjuice, *S. officinalis*, and *A. vulgaris* showed dose-dependent cytotoxic effects in both cell lines, as shown in Figures 1 and 2. At doses less than 5443 µg/mL, 613 µg/mL, and 805.7 µg/mL for Verjuice, *S. officinalis*, and *A. vulgaris*, respectively, there was no evidence of cytotoxicity in macrophages derived from THP-1 cells. There was also no detectable reduction in cell viability in 3T3-L1 cells exposed to doses of 795.2 µg/mL, 4820 µg/mL, and 5079 µg/mL Verjuice and *S. officinalis*, respectively. Doses of 795.2 µg/mL, 4820 µg/mL, and 5079 µg/mL showed no detectable decrease in cell viability. Table 1 summarizes the IC50 values measured in both cell cultures.

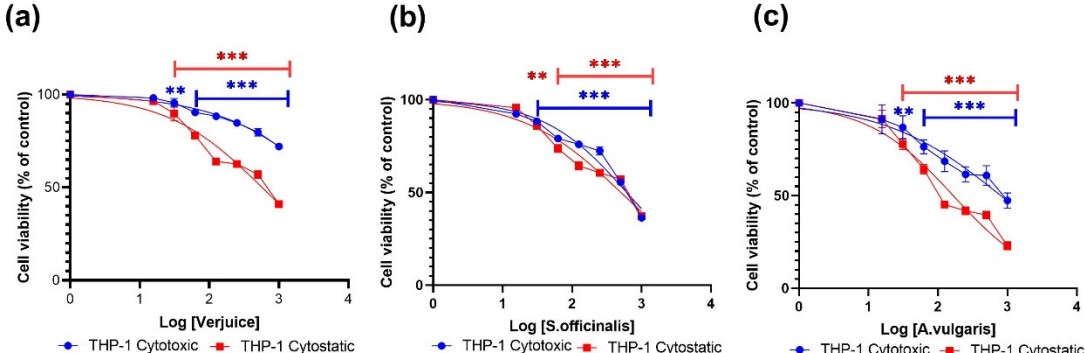

**Figure 1.** Displays the findings of the MTT experiment for the cytotoxic and cytostatic effects of THP-1-derived macrophages. Cells treated with various plant extract concentrations (1000 μg/mL–0 μg/mL (control): (**a**) Verjuice, (**b**) *S. officinalis*, and (**c**) *A. vulgaris*. Extract concentration expressed as Log [concentration] against normalized optical density with 100% set as the control value. Data represents the mean of three independent experiments conducted in triplicate. The asterisk indicates statistically significant difference from control cells which was calculated using a one-way ANOVA followed by Dunnett's multiple comparisons test. (** $p < 0.01$ vs. LPS, and *** $p < 0.001$ vs. LPS).

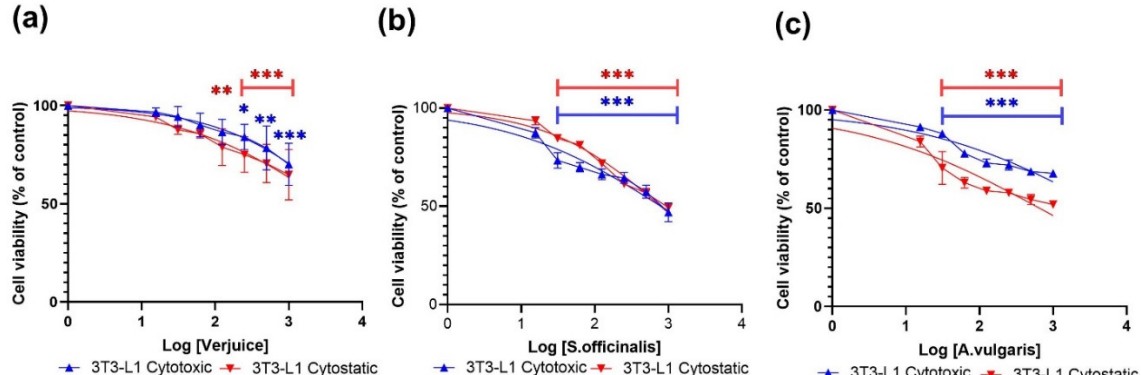

**Figure 2.** Shows the cytotoxic and cytostatic effects of various doses of plant extracts 1000 μg/mL–0 μg/mL (control) on 3T3L-1 cells. (**a**) Verjuice, (**b**) *S. officinalis*, and (**c**) *A. vulgaris*. Extract concentration expressed as Log [concentration] against normalized optical density with 100% set as the control value. Data represents the mean of three independent experiments conducted in triplicate. The asterisk indicates statistically significant difference from control cells which was calculated using a one-way ANOVA followed by Dunnett's multiple comparisons test. (* $p < 0.05$ vs. LPS, ** $p < 0.01$ vs. LPS, and *** $p < 0.001$ vs. LPS).

**Table 1.** IC$_{50}$ values of test plant extracts in THP-1 and 3T3-L1 cell lines. The IC$_{50}$ values were determined by fitting the dose–response curves to non-linear regression curve fit using GraphPad PRISM 8.

| Extract | Cytotoxic μg/mL | | Cytostatic μg/mL | |
|---|---|---|---|---|
| | **THP-1** | **3T3-L1** | **THP-1** | **3T3-L1** |
| **Verjuice** | 5443 | 5079 | 574.6 | 3486 |
| *S. officinalis* | 613.0 | 795.2 | 505.0 | 787.8 |
| *A. vulgaris* | 805.7 | 4820 | 166.1 | 660.0 |

As for the cytotoxic effects, the antiproliferative effects of the three extracts were evaluated using the MTT assay. Figures 1 and 2 show that all three extracts exhibit cyto-

static effects in a dose-dependent manner. The effects were significantly higher in cells for the THP-1-derived macrophages compared to the effects observed in 3T3-L1. The $IC_{50}$ values for THP-1-derived macrophages were 574 µg/mL, 505 µg/mL, and 166 µg/mL for Verjuice, *S. officinalis,* and *A. vulgaris*, respectively (Table 1). Among the three extracts tested, *A. vulgaris* was found to induce significant antiproliferative activity at non-toxic concentrations. More research and study are needed to understand the underlying processes that cause this impact. Extracts from plants including *Pinus pinaster*, *maritime pine* tree, *Ginkgo biloba*, grape seed extract, green tea, *Alchemilla vulgaris*, *Olea europea*, and *Mentha longifolia* L., as well as seeds of *Cuminum cyminum*, have previously been stated to suppress adipogenesis in adipocytes [34–37].

### 3.2. IL-6, TNF-α, IL-10, and NO

The anti-inflammatory effects of the three extracts were analyzed in LPS-activated THP-1-derived macrophages. The levels of TNF-α, IL-6, IL-10, and NO were evaluated by using enzyme-linked immunosorbent assay, and Griess assay. We found an increasing levels of TNF-α, IL-6, IL-10, and NO post LPS treatment (Figures 3–5).

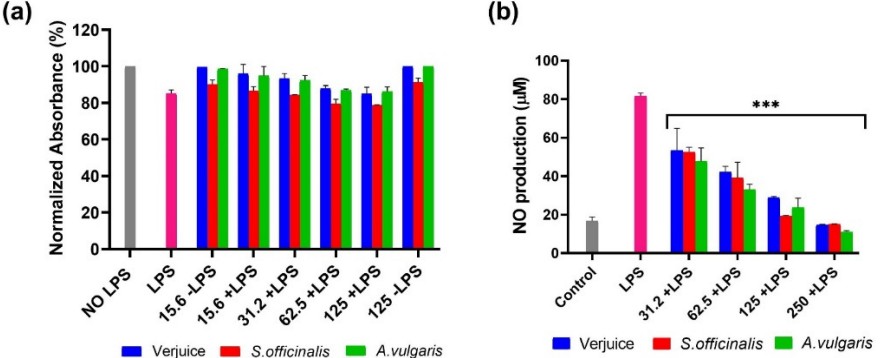

**Figure 3.** Shows the cell viability (**a**) and nitric oxide (NO) production by LPS-activated THP-1-derived macrophages (**b**) following a 72 h incubation with various plant extract concentrations (125 µg/mL–15.6 µg/mL). The asterisk indicates statistically significant difference from LPS-treated cells which was calculated using a one-way ANOVA followed by Dunnett's multiple comparisons test. (*** $p < 0.001$ vs. LPS).

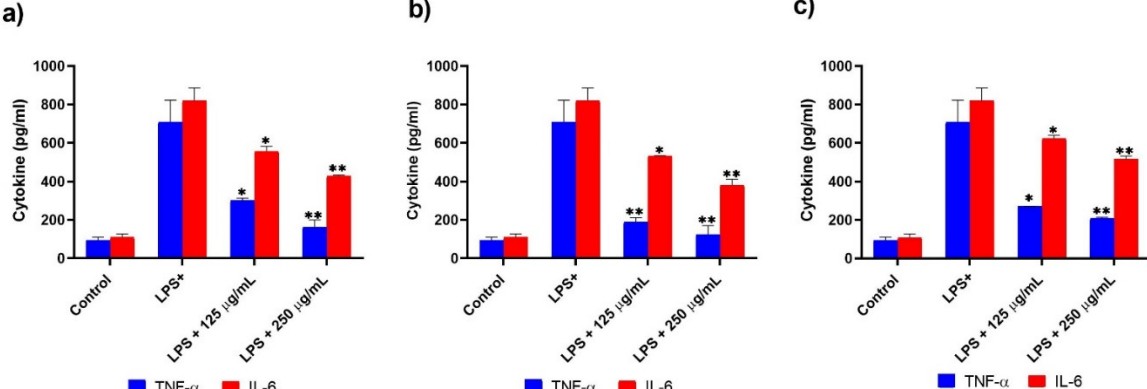

**Figure 4.** Effect of verjuice (**a**), *S. officinalis* (**b**), and *A. Vulgaris* (**c**) extracts (250 µg/mL and 125 µg/mL) on LPS-induced THP-1-derived macrophages on pro-inflammatory cytokines TNF-α and IL-6 production (pg/mL) after a 4 h and 6 h incubation, respectively, with two particular doses. Each bar represents the mean ± SD of three different experiments carried out in duplicate. The asterisk indicates statistically significant difference from LPS treated cells which was calculated using a one-way ANOVA followed by Dunnett's multiple comparisons test. (* $p < 0.05$ vs. LPS, ** $p < 0.01$ vs. LPS).

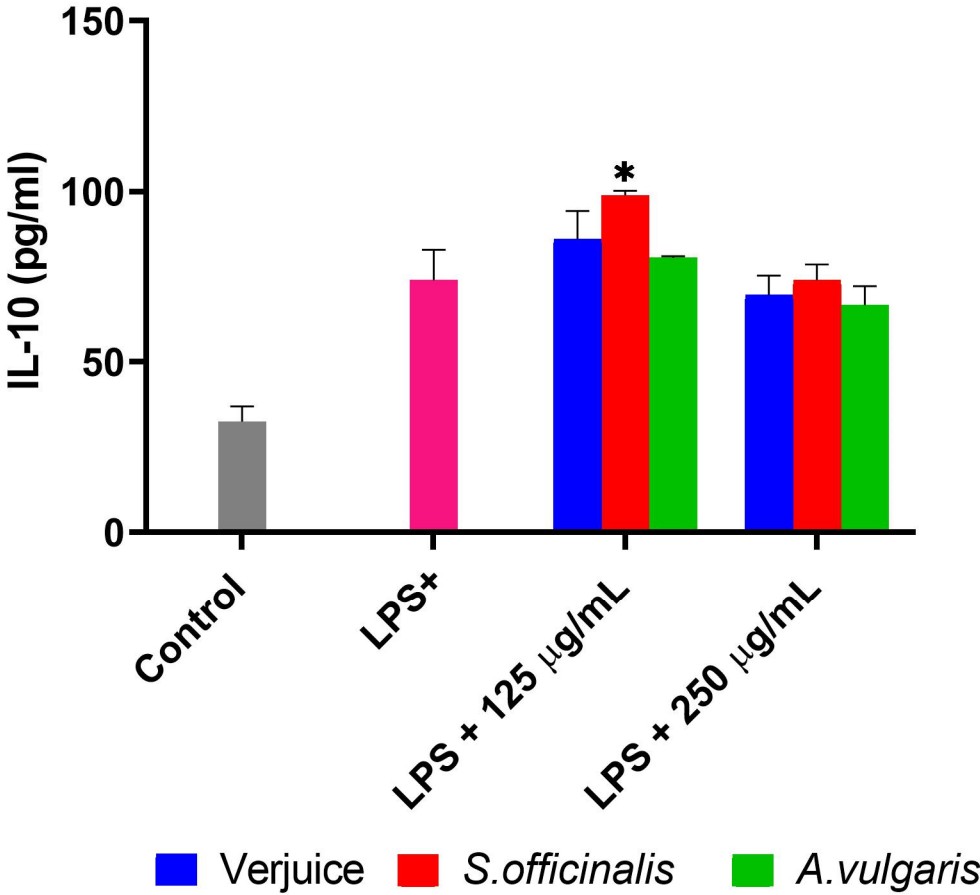

**Figure 5.** Shows the impact of extracts from verjuice, *S. officinalis*, and *A. vulgaris* on LPS-induced THP-1-derived macrophages on IL-10 production (pg/mL) following a 24 h incubation with extract concentrations of 250 µg/mL and 125 µg/mL. Each bar indicates the mean ± SD of three independent experiments carried out in triplicate. A one-way ANOVA was used to quantify the statistical difference from LPS-treated cells, and Dunnett's multiple comparison test was used to determine the difference. (* $p = 0.05$ vs. LPS).

Figure 3 represents the dose-dependent inhibition of the LPS-activated production of NO by verjuice, *S. officinalis*, and *A. vulgaris* extracts in THP-1-derived macrophages at non-toxic concentrations. The NO reached control levels of untreated cells (16.60 ± 2 µM). Similar results were reported with a large number of medicinal plants, crude extracts, and phytochemicals [38–42]. For example, *Hypericum triquetrifolium* (HT) extracts were reported to suppress the LPS-induced NO production through attenuation of the LPS-induced transcription of iNOS [39]. The action mechanism by which verjuice, *S. officinalis*, and *A. vulgaris* extracts inhibit the NO secretion needs to be evaluated in future experiments. These plants were reported to contain various potential antioxidant and anti-inflammatory compounds. Verjuice is rich in catechin and anthocyanin which are also found in grape juice. Moreover, verjuice is an important source of bioactive compounds such as phenolic, hydroxybenzoic, hydroxycinnamic, caffeic, caftaric, fertaric, gallic, p-coumaric, protocatechuic acids, as well as catechin, epicatechin, quercetin-glucoside, and tyrosol [20]. While fractionated methanolic extract from *S. officinalis* leaves contain diterpenes (carnosic acid, carnosol, royleanonic acid, and 7-methoxyrosmanol) in addition to triterpenes (oleanolic acid), rutin, quercetin, and salvianolic acids which make *S. officinalis* exhibit anti-hyperglycemic, anti-hyperlipidemic, anti-inflammatory, antioxidant, pancreatic lipase and lipid absorption inhibition, lipid peroxidation inhibition, PPARγ agonistic activity and metformin-like effects [43–46]. *A. vulgaris* extract is rich with gallic acid, caffeic acid, catchin, and quercetin,

and an investigation of *A. vulgaris* extract showed that it exhibits high antioxidant and anticancer properties, and leads to the inhibition of enzymatic activities [47,48].

The levels of pro-inflammatory (IL-6 and TNF-$\alpha$) and anti-inflammatory (IL-10) cytokines produced by LPS-activated THP-1-derived macrophages in the culture medium were evaluated using a commercial ELISA kit. TNF-$\alpha$ and IL-6 production peaked at 4 and 6 h after LPS stimulation, respectively. TNF-$\alpha$ and IL-6 levels were dramatically suppressed by test plant extracts in a dose-dependent manner, as shown in Figure 4a,b. *S. officinalis* decreased the generation of TNF-$\alpha$ and IL-6 at 250 pg/mL concentrations by 84% and 55%, respectively, when compared to control LPS-treated cells.

In contrast to the inhibitory effects on TNF-$\alpha$ and IL-6, the anti-inflammatory cytokine IL-10 was elevated following LPS-activated THP-1-derived macrophage treatment. Although untreated cells produced small quantities of IL-10, TNF-$\alpha$, and IL-6, *S. officinalis* extract inhibited their production significantly, albeit at a lower dose of 125 µg/mL. When compared to untreated cells, *S. officinalis* extract produced much more IL-10 (98.90 ± 1.3 pg/mL) than LPS-treated cells (73.90 ± 8.9 pg/mL) as shown in Figure 5.

### 3.3. Effects of the Plant Extracts on Adipogenicity

Many cell lines have recently been found to undergo in vitro lipogenic differentiation into adipocytes. Among them, the 3T3-L1 preadipocyte is a well-studied biological model for understanding the adipogenesis process. After being stimulated by a chemical cocktail, 3T3-L1 preadipocytes develop into mature adipocytes, with triglyceride (TG) accumulation being one of the hallmarks of adipogenesis [2–4]. To assess the effect of test plant extracts on 3T3-L1 cells adipogenesis, differentiation of 3T3-L1 pre-adipocytes to mature adipocytes was induced in the presence of different concentrations of the test plants (0–2 days) and the treatment was sustained for a total 10 days. The fat accumulation was assessed in fully differentiated adipocytes, which were treated with different concentrations of the test plants from day 10 to day 15 of differentiation for a period of 5 days. Adipogenesis and lipid accumulation were evaluated by Oil Red O staining using an optimized protocol [49]. Cells were initially examined and photographed under a microscope, after which they were eluted in 2-propanol and the optical density values were determined using an ELISA reader. To measure intracellular triglyceride buildup, data were normalized and computed as a percentage of the control (%). In addition, $ID_{50}$ values (Table 2) were calculated for the test plant extracts to determine the reduction of cell adipogenesis and lipid accumulation at half-maximal levels (50%), which was calculated by graphic interpolation of the dose–response curves to non-linear regression curve fit using GraphPad PRISM 8, as shown in Figure 6.

**Table 2.** $ID_{50}$ values of test plant extract concentrations at which the reduction of the cell adipogenesis and lipid accumulation at the half maximal level 50% was determined by graphic interpolation of the dose–response curves to non-linear regression curve fit using GraphPad PRISM 8.

| Extracts | $ID_{50}$ Adipogenesis µg/mL | $ID_{50}$ Lipid Accumulation µg/mL |
|---|---|---|
| Verjuice | 57.84 | 155.6 |
| *S. officinalis* | 40.18 | 61.89 |
| *A. vulgaris* | 183.8 | 148.4 |

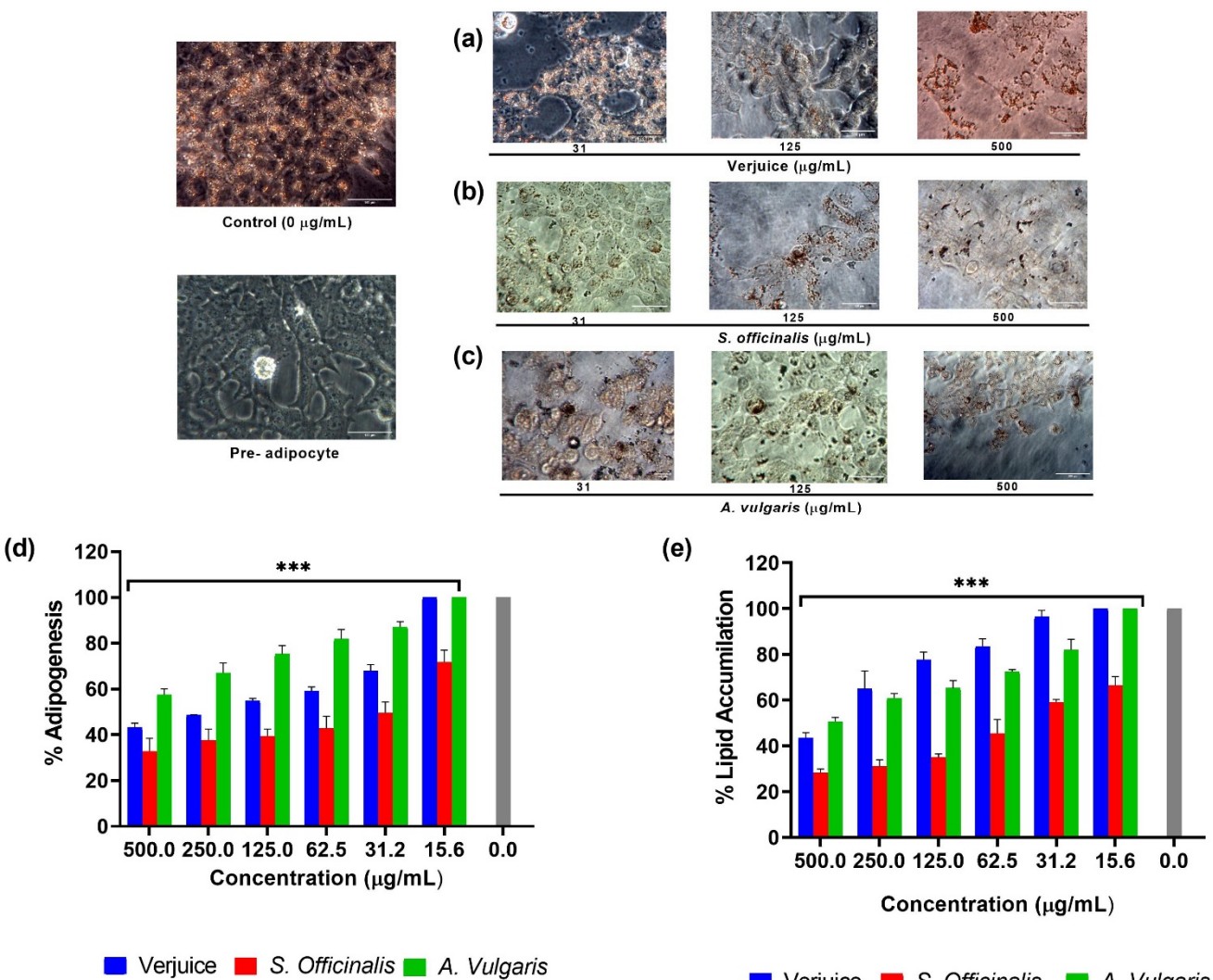

**Figure 6.** (**a–c**) Shows how different extracts (31, 125, and 500 μg/mL) from verjuice, *S. officinalis*, and *A. vulgaris* affect the amount of fat that accumulates in 3T3-L1-derived differentiated adipocytes. (**d**) The relative density of the lipid contents in 3T3-L1-derived differentiated adipocytes exposed to various extract concentrations (500 μg/mL to 0 μg/mL). (**e**) The relative density of lipid contents in completely developed adipocytes that were exposed to various extract concentrations for 5 days (days 10 to 15) (500 μg/mL to 0 μg/mL). Each bar represents the mean ± SD of three different experiments carried out in triplicate. The asterisk indicates statistically significant difference from control (untreated cells) which was calculated using a one-way ANOVA followed by Dunnett's multiple comparisons test. (*** $p < 0.001$ vs. ctrl).

Microscopic examination of 3T3-L1 cells stained with Oil Red O after exposure to different concentrations of tested plant extract (31, 125, and 500 μg/mL) revealed that the differentiation process as well as intracellular triglyceride accumulation produced a dose-dependent effect in the number of fully differentiated cells with accumulated lipid droplets. Figure 6a–c shows that there was no cytotoxicity whenever the morphology of control and treated 3T3-L1 was compared. As the dosages rose, there was a dramatic reduction in cytoplasmic lipid droplet formation with distinct intracellular morphology with smaller lipid droplets or even none at all. These findings were supported by eluting the dye and measuring the optical density following microscopic examination (Figure 6).

When compared to the control group of differentiated cells with 0.523 ± 0.02 average absorbance, the average absorbance of undifferentiated pre-adipocyte media was 0.202 ± 0.09, suggesting that lipid accumulation rose by 258.91% and differentiation pro-

gressed effectively. As indicated in Figure 6, lipid buildup in the test plants group was dramatically reduced in all samples in a dose-dependent manner. The ID50 for test plants is shown in Table 2, indicating that S. officinalis has the greatest effect on differentiation inhibition and lipid accumulation suppression at nontoxic concentrations of 40.18 μg/mL and 61.89 μg/mL, respectively.

Previously, extracts from some plants, such as the extracts of grape seeds, green tea, *Allium sativum*, *Camellia sinensis*, *Capsicum annuum*, *Curcuma longa*, *Ginkgo biloba*, *Olea europea,* and *Mentha longifolia* [37,38] have been reported to reduce adipogenesis in adipocytes. In a previous study, we have shown that the leaves of *S. officinalis* act via synergistic mechanisms to reduce the weight of overweight and obese patients [28,35]. As a result, more research should be conducted to see if these extracts may be combined to form a synergistic composition that inhibits adipogenesis more effectively than the separate extracts. Furthermore, during adipogenesis, adipogenic signals govern adipocyte differentiation and intracellular fat deposition by activating transcriptional activators, mostly from the PPAR and C/EBP families. [2–4]. These two nuclear factors synchronize the complex operation of adipogenic gene expression throughout terminal preadipocyte differentiation by triggering the expression of multiple adipogenic gene products, such as ADRP, aP2, CD36, and perilipin, all of which work together to accomplish the adipocyte phenotype [2–4]. The role of these transcription factors in our observed results will be investigated in future studies.

## 4. Conclusions

The production of pro-inflammatory and anti-inflammatory cytokines, as well as adipocyte differentiation and fat accumulation in the 3T3-L1 mouse embryo fibroblast cell line and the human monocytic cell line THP-1-derived macrophages, were measured to determine the anti-inflammatory and antiadipogenic effects of ethanolic extracts of verjuice *Vitis vinifera* L., *Salvia officinalis* L., and *Alchemilla vulgaris* L. On both cell lines, the three extracts exhibited cytostatic effects at non-cytotoxic concentrations. During treatment with 125 μg/mL, the three extracts dramatically reduced the LPS-induced NO, TNF-α, and IL-6 production in a dose-dependent manner with the highest effects being reached at 250 μg/mL. In addition, treatment with plant extracts increased the production levels of IL-10. The adipocyte differentiation and fat accumulation in 3T3-L1 were decreased to 20% of control values after treatment with the plant extracts. Taken together, these results suggest that *V. vinifera*, *S. officinalis*, and *A. vulgaris* likely exert their anti-obesity effects through cytostatic actions and modulation of pro-inflammatory and anti-inflammatory cytokine production, as well as by reducing adipocyte differentiation and fat accumulation.

Based on published data, the differentiation and lipid accumulation inhibition in 3T3-L1 cells together with pro-inflammatory cytokine suppression results seem to be associated with the polyphenolic content of the test plant extracts. Further confirmatory experiments at the biochemical and molecular level are encouraged to be done in the future.

**Author Contributions:** B.M., N.S., A.K., N.H., S.S., O.S., R.Z. and B.S. contributed to the study conception and design. The first draft of the manuscript was written by B.M., A.K. and B.S., B.M., N.S., A.K., N.H., S.S., O.S., R.Z. and B.S. commented on previous versions of the manuscript. B.M., N.S., A.K., N.H., S.S., O.S., R.Z. and B.S. All authors have read and agreed to the published version of the manuscript.

**Funding:** This research received no external funding.

**Institutional Review Board Statement:** Not applicable.

**Informed Consent Statement:** Not applicable.

**Data Availability Statement:** The human monocytic cell line THP-1 (ATCC 202-TIB) and the mouse embryo fibroblasts cell line 3T3-L1 (ATCC CL-173) were purchased from American Type Culture Collection (Manassas, VA, USA).

**Conflicts of Interest:** The authors declare no conflict of interest.

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
