# Peer review of "Anti-Inflammatory and Anti-Adipogenesis Effects of Alchemilla vulgaris L., Salvia officinalis L., and Vitis vinifera L. in THP-1-Derived Macrophages and 3T3-L1 Cell Line"

_2673-5601, doi:10.3390/immuno3020010_

Round 1
Reviewer 1 Report
The idea of testing new 3 extracts on adipogenesis is great, but the formatting of the manuscript is not good. The way of presentation and analysis of the data is not appropriate and the figure legend is mixed up with the contents which should be in the result and discussion section. Overall, the writing is poor and needs to be carefully improved intensively.
Author Response
Thank you very much for your valuable comments. We have revised the manuscript according to all addressed comments and suggestions. We are also ready to revise again if you have other requests and/or suggestions.
The way of presentation and analysis of the data is not appropriate and the figure legend is mixed up with the contents which should be in the result and discussion
section.
- All Figure legends were corrected.
- We replaced Figure 3 with a new one.

Reviewer 2 Report
Reviewer comments:
The present study focuses on the anti-inflammatory and anti-adipogenicity effects of the herbal extracts of medicinal plants, in THP-1 and 3T3-L1 cell lines. The extracts exert their cytostatic effects and modulation of the assembly of pro-inflammatory and anti-inflammatory cytokines TNF-alpha and IL-6), as well as through reduction of adipocytes differentiation and fat accumulation by 20%.
The queries/suggestions that needs to be addressed are:
1. In line 91, section material and methods, “tape water” should be written as “tap water”.
2. Was there any time kinetics followed while determining the cytotoxicity? (12-hour, 24 hour, 48-hour, 72 hour) before determining the activities imparted by the extracts.
3. What was the positive control used while performing anti-adipogenicity studies?
4. In line 137, was 50 liters of supernatant and 200 liters of Griess reagent used?
5. In figure 4, the data shown is post 4-hour stimulation or 6 hours? Be precise with the time points in the figure legend.
6. One of the major concerns with plant extracts is their ability to induce hemolysis after administration. Was that concern addressed by performing hemolytic assay on red blood cells? If done then it should be included in the study and if not then it should be performed and incorporated.
7. iNOS and COX-2 are upregulated by LPS treatment, it would be interesting to show their downregulation in the herbal extract treated groups by immunoblotting.
Author Response
Thank you very much for your valuable comments. We have revised the manuscript according to all addressed comments and suggestions. We are also ready to revise again if you have other requests and/or suggestions.
In details:
- In line 91, section material and methods, “tape water” should be written as “tap water”.:
Corrected
- Was there any time kinetics followed while determining the cytotoxicity? (12-hour, 24 hour, 48-hour, 72 hour) before determining the activities imparted by the extracts.
Based on procedure published data in numerous scientific papers, the cytotoxic effects are routinely measured in confluent cell cultures after 24h of treatment. While the cytostatic effects are measured at low cell density (cell proliferation) after 72h. Therefore, it is very difficult to assess cytotoxic effects after 72h, because the cell will de-attach from the 96-well plate
- What was the positive control used while performing anti-adipogenicity studies?
The untreated cells were taken as a positive control. These cells were maintained in “Maintenance Medium” supplemented with insulin
- In line 137, was 50 liters of supernatant and 200 liters of Griess reagent used?
Corrected
- In figure 4, the data shown is post-4-hour stimulation or 6 hours? Be precise with the time points in the figure legend.
Corrected: we measured the TNF-alpha after four hours and the IL-6 after six hours.
- One of the major concerns with plant extracts is their ability to induce hemolysis after administration. Was that concern addressed by performing hemolytic assay on red blood cells? If done then it should be included in the study and if not then it should be performed and incorporated.
The used plants are used in traditional medicine as tees. Therefore, we applied water/ethanoic extracts. We think that hemolytic tests at this early phase are not necessary. We will do these tests in our future tests with isolated active compounds.
- iNOS and COX-2 are upregulated by LPS treatment, it would be interesting to show their downregulation in the herbal extract treated groups by immunoblotting.
Yes, it is well documented that these enzymes are upregulated by LPS. We have shown in other papers with other plants that both enzymes are downregulated at transcript levels
Saad B, Soudah AbouAtta B, Basha W, Kmeel A, Khasib S, Hmade A, & Said O, (2008) Hypericum triquetrifolium-derived factors down regulate the production levels of nitric oxide and pro-inflammatory cytokine TNF in LPS-Activated THP-1 cells. Evidence based complementary and alternative medicine doi:10.1093/ecam/nen056
Saad B, Embaslat W, Abu Farich B, Mahajna Sh, Azab M, Daragmeh J, Khasib S, & Zaid H, (2016) Hypericum Triquetrifolium –extracts modulate IL-6, IL-10 and TNF-a protein and mRNA expression in LPS-activated human peripheral blood mononuclear cells and THP-1-derived macrophages, Medicinal & Aromatic Plants, doi:10.4172/2167-0412.S3-004
However, since these measurements are time-consuming, it will be very difficult for us to design and carry out these experiments within the 10-days period given by the journal

Reviewer 3 Report
The authors have attempted to test the anti-inflammatory and anti-adipogenesis effects of 3 medicinal plant extracts. The authors provide a reasonable introduction and background to why this research question needs to be addressed, however, they do not specify the reasons for use of specific doses for these extracts, making it a bit confusing to understand where the numbers are derived from. Reporting of methods, results and statistics needs to be improved significantly. For example, Fig3a,b,c combine results of two separate, unrelated read outs (NO production and MTT) making them difficult to interpret. It is unclear which groups were compared to perform the statistics. Please also check reporting of concentrations and scientific notations in the methods section (lines 135, 137). The conclusions and discussion section need to be revised as it is highly inadequate in its current form. The authors first need to summarize their findings in both THP-1 and 3T3 cell lines and then provide concluding remarks on goals for future studies
Author Response
Thank you very much for your valuable comments. We have revised the manuscript according to all addressed comments and suggestions. We are also ready to revise again if you have other requests and/or suggestions.
In details:
The authors have attempted to test the anti-inflammatory and anti-adipogenesis effects of 3 medicinal plant extracts. The authors provide a reasonable introduction and background to why this research question needs to be addressed, however, they do not specify the reasons for use of specific doses for these extracts, making it a bit confusing to understand where the numbers are derived from.
In the first phase of our experiments, we carried out a cytotoxic test to assess the safe and non-toxic concentrations of the plant extracts to be used in further experiments. Based on cytotoxic results (MTT assay) we determined the concentration used in the second phase, in which we evaluated the anti-inflammatory and adipogenesis. In general, plant extracts are used in the range 0-1mg/ml.
Saad B, Azaizeh H , Abu Hijleh G , & Said O , (2006) Safety of traditional Arab herbal medicine. Evidence based complementary and alternative medicine 3:433-439.
Reporting of methods, results and statistics needs to be improved significantly. For example, Fig3a,b,c combine results of two separate, unrelated read outs (NO production and MTT) making them difficult to interpret.
We replaced the Figure with a new version.
It is unclear which groups were compared to perform the statistics.
We replaced the Figure with a new version.
Please also check reporting of concentrations and scientific notations in the methods section (lines 135, 137).
Corrected
The conclusions and discussion section need to be revised as it is highly inadequate in its current form.
The authors first need to summarize their findings in both THP-1 and 3T3 cell lines and then provide concluding remarks on goals for future studies
The valuable comments of the reviewer are included in the rewritten conclusions

Round 2
Reviewer 1 Report
Well done. Now the manuscript is satisfied.
Author Response
We thank the reviewer for his valuable comments

Reviewer 2 Report
The answers provided by authors are satisfactory and well accepted.
Author Response

(The authors gave the same response as above.)

Reviewer 3 Report
The authors have made improvements to the text but the revised figures still do not portray the comparisons and statistics accurately. For all figures and questions, comparisons are made between LPS/ control vs individual plant extract. However, the graphs are all grouped by plant extracts being plotted together. The graphs should instead be separated for LPS/ control with each plant extract. Also, it is unclear why LPS condition is the exact same value for each extract. Either the LPS condition was performed for each plate with individual plant extracts, in which case, it is biologically impossible that all plates had the exact same reading of absorbance or concentration OR there was only one LPS condition well, in which case, there need not be three bars. The data need to be represented for scientific accuracy as the aim of this study is not directed towards comparison between the three extracts but rather with controls.
Author Response
We thank also the reviewer for your valuable comments. We have revised the manuscript according to all addressed comments and suggestions.
Best regards.
Prof Bashar Saad
Point by point:
The authors have made improvements to the text but the revised figures still do not portray the comparisons and statistics accurately.
For all figures and questions, comparisons are made between LPS/ control vs individual plant extract. However, the graphs are all grouped by plant extracts being plotted together.
We redraw Figures 3-6 according to your valuable comments
The graphs should instead be separated for LPS/ control with each plant extract.
Corrected in all Figures
Also, it is unclear why LPS condition is the exact same value for each extract. Either the LPS condition was performed for each plate with individual plant extracts, in which case, it is biologically impossible that all plates had the exact same reading of absorbance or concentration OR there was only one LPS condition well, in which case, there need not be three bars.
Corrected in all Figures
The data need to be represented for scientific accuracy as the aim of this study is not directed towards comparison between the three extracts but rather with controls.
We revised Figures 3-6 according to your valuable comments

Round 3
Reviewer 3 Report
The authors have improved language, description of methods and conclusions significantly. The authors have addressed my comment regarding specific figures (used as examples) but have disregarded the general application of those to all figures. For example, in Fig 4, the comparison and stats between LPS and individual cytokines (TNFa or IL-6). The graph needs to depict the groups being compared as one set i.e all blue bars of LPS with blue bars for LPS+ plant extract in panels a,b and c and same for red bars. The statistics should then be denoted on top on the bars being compared with the relevant * for p value. The figures require reformatting to abide by this basic concept of accuracy in data reporting. Additionally, the MTT assay results are typically reported as OD readings at 570nm or % cell viability. Please consider performing % cell viability calculations and perform appropriate statistical analysis for the same.
Author Response
We thank also the reviewer for your valuable comments.
The authors have improved language, description of methods and conclusions significantly.
Many thanks, this is the result of your valuable suggestions
The authors have addressed my comment regarding specific figures (used as examples) but have disregarded the general application of those to all figures.
Yes we have changed the Figures according to your suggestions
For example, in Fig 4, the comparison and stats between LPS and individual cytokines (TNFa or IL-6). The graph needs to depict the groups being compared as one set i.e all blue bars of LPS with blue bars for LPS+ plant extract in panels a,b and c and same for red bars. The statistics should then be denoted on top on the bars being compared with the relevant * for p value. The figures require reformatting to abide by this basic concept of accuracy in data reporting.
We disagree with the respected reviewer, we think that the current style is clear and easy to understand. If we group the TNFalpha and IL-6, we will have 6 Figures instead of 3 !!!. We have published such Figures in more than 20 publications published in journals with relatively high impact factor.
Additionally, the MTT assay results are typically reported as OD readings at 570nm or % cell viability. Please consider performing % cell viability calculations and perform appropriate statistical analysis for the same. The presented data are % control as mentioned in the Figure legends
We redrew the Figures according to your valuable comments
We hope that the revised manuscript is now fit for publication in Immuno
